# High diversity of fungal ecological groups from ice-free pristine and disturbed areas in the Fildes Peninsula, King George Island, Antarctica

**Sigisfredo Garnica[1]\*, Genaro Soto-Rauch[1], Ellen M. Leffler[2], Christian Núñez[3], Jonhatan Gómez-Espinoza[4], Enzo Romero[3], Ramón Ahumada-Rudolph[3], Jaime R. Cabrera-Pardo[3,5]\***

1 Facultad de Ciencias, Instituto de Bioquímica y Microbiología, Universidad Austral de Chile, Isla Teja, Valdivia, Chile, 2 Department of Human Genetics, The University of Utah School of Medicine, Salt Lake City, UT, United States of America, 3 Departamento de Química, Laboratorio de Química Aplicada y Sustentable (LabQAS), Universidad del Bío-Bío, Concepción, Chile, 4 Departamento de Ciencias, Liceo Técnico Profesional Diego Portales, Linares, Chile, 5 College of Dental Medicine, Roseman University of Health Sciences, South Jordan, UT, United States of America

\* jacabrera@ubiobio.cl (JRC-P); sigisfredo.garnica@uach.cl (SG)

**Data Availability Statement:** Raw sequencing data files are available from the Sequence Read Archive

## Abstract

Ice-free areas are habitats for most of Antarctica's terrestrial biodiversity. Although fungal communities are an important element of these habitats, knowledge of their assemblages and ecological functions is still limited. Herein, we investigated the diversity, composition, and ecological functionality of fungal communities inhabiting sediments from ice-free areas across pristine and anthropogenically impacted sites in the Fildes Peninsula on King George Island, Antarctica. Samples were collected from both pristine and disturbed areas. We used the internal transcribed spacer (ITS1) region via Illumina sequencing of 34 sediment samples for fungal identification. The Ascomycota (14.6%) and Chytridiomycota (11.8%) were the most dominant phyla, followed by Basidiomycota (8.1%), Rozellomycota (7.0%), Mucoromycota (4.0%), while 34.9% of the fungal diversity remained unidentified. From a total of 1073 OTUs, 532 OTUs corresponded to 114 fungal taxa at the genus level, and 541 OTUs remained unassigned taxonomically. The highest diversity, with 18 genera, was detected at site A-3. At the genus level, there was no preference for either pristine or disturbed sites. The most widely distributed genera were *Betamyces* (Chytridiomycota), occurring in 29 of the 34 sites, and *Thelebolus* (Ascomycota), detected in 8 pristine sites and 7 disturbed sites. The *Glomeraceae* gen. incertae sedis was more common in disturbed sites. A total of 23 different ecological guilds were recorded, with the most abundant guilds being undefined saprotrophs, plant pathogens, plant saprotrophs, pollen saprotrophs, and endophytes. The fungal communities did not show significant differences between pristine and disturbed sites, suggesting that the anthropogenic impact is either not too intense or prolonged, that the spatial distance between the sampled sites is small, and/or that the environmental factors are similar. Although our study revealed a high fungal diversity with various ecological specializations within these communities, nearly one-third of the diversity could not be

(SRA) under BioProject number PRJNA1181963 (http://www.ncbi.nlm.nih.gov/bioproject/1181963).

**Funding:** Instituto Antártico Chileno (INACH), grant INACH RT_16-21. J.R.C-P and VRIP - Universidad del Bio Bio, grants GI2310643 and EQ2326450.

**Competing interests:** The authors have declared that no competing interests exist.

assigned to any specific taxonomic category. These findings highlight the need for further taxonomic research on fungal species inhabiting ice-free areas. Without identifying the species present, it is difficult to assess potential biodiversity loss due to environmental changes and/or human activities.

## Introduction

Antarctica, where temperatures can drop well below -40 degrees Celsius and even lower in some inland areas, is one of the most extreme places on our planet [1]. Coastal regions, however, are milder, with summer temperatures typically ranging from -2 to 8 degrees Celsius. Because of its unique environment, ecosystems, and geographic isolation, Antarctica is an ideal location for research into climate change, biological invasions, geology, and biodiversity [2]. Biota dispersal opportunities between ice-free areas are severely limited by physical and geographical barriers [3].

Previous investigations have shown that most of Antarctica's biodiversity is found almost exclusively in ice-free areas [4]. These areas comprise less than 0.5% of the continent [5] and harbor diverse communities of microbes, animals, and plants (e.g., [6]). Research on Antarctic fungi has primarily focused on understanding the diversity, distribution, and ecology of these organisms under extreme conditions. Studies have explored fungal diversity using pure culture isolates from soils [7–9], mosses [10,11], water columns from permanently ice-covered lakes [12], permafrost and active layers [13], snow [14], thermal soil gradients [15], lake sediments [9], marine macroalgae [16], and vascular plants [17,18].

Moreover, fungal isolations from different Antarctic substrates have been conducted to investigate diversity and potential biotechnological applications, such as the production of pigments, enzymes, and bioactive compounds [19–26]. Other authors have investigated by PCR amplification and cloning the diversity of fungal communities from soils [27], lake water [28] or using next-generation sequencing (NGS) analyses from marine sediments [29], salt encrustation in rocks [30], lake water [31–33], lake sediments [33–38], soils [39–43], air and snow [14,44–46] and different substrates from ice-free areas [6]. The use of Sanger and NGS methods have enabled the rapid identification of numerous microorganisms from various samples [see e.g. 47]. Some studies have found that certain fungi are endemic to Antarctica, meaning they are exclusively from this region [13,17,28]. Other recent research has sought to understand how fungal communities respond to new conditions and the impacts of human activities, such as tourism and scientific research stations, on Antarctic fungi [15,40].

Over the past 50 years, human activities in Antarctica, particularly in the Antarctic Peninsula where scientific stations and shelters are located, have increased. These activities have caused various pollution issues, including fuel combustion, accidental oil spills, waste incineration, sewage disposal, and other station-related impacts (e.g., [23,48]). The Fildes Peninsula on King George Island, which hosts six permanent research stations, serves as a major hub for scientific, logistical, military, and tourism activities. Research station operations and tourism may introduce non-native fungal species to Antarctica [23,49,50]. Previous research indicates that anthropogenic activities have facilitated the spread of cosmopolitan fungal species. Understanding the impact of these activities is crucial to develop strategies that minimize their effects on endemic species in this sensitive environment (e.g., [2]).

In the Fildes Peninsula, extreme climatic conditions and abiotic factors, such as temperature fluctuations, ultraviolet radiation, and glacier melt, can influence fungal composition

[41,51]. Additionally, the simplified community structure of the organisms—comprising few primary producers and consumers—offers a unique opportunity to define the parameters of microbial colonization and interactions [52]. Human activities, including tourism and scientific research, are carefully regulated to minimize environmental impact and preserve unique ecosystems. However, studies characterizing the diversity and ecological specializations of fungal communities in these extreme sites and their links to human impact remain scarce.

This research, conducted in the Fildes Peninsula on King George Island, used Illumina sequencing to reveal: (1) the diversity and composition of fungal communities inhabiting sediments from both pristine and human-disturbed ice-free areas; (2) the ecological functionality of fungal communities across these sites; and (3) the impact of human activities on fungal community composition. We hypothesized that fungal communities in ice-free sediments would differ between sites exposed to tourism and scientific activities and those without such impacts. Our research aimed to contribute to the knowledge of fungal communities in coastal marine sediments from both impacted and non-impacted ice-free areas. The findings will enhance our understanding of true fungal diversity in these ice-free oases and enable accurate monitoring of human activity impacts, supporting conservation plans in this fragile environment.

## Material and methods

### Ethics statement

The material used in this study is protected; therefore, a specific permit was requested for sampling. Sampling was conducted with the permission of the Instituto Antárctico Chileno–Ministerio de Relaciones Exteriores, Chile, Certificate No. 369.

### Study sites

In March 2023, the 59th Scientific Antarctic Expedition (ECA59) of INACH took place at the Julio Escudero Base (INACH), located on the Fildes Peninsula, King George Island, Antarctica. Glacial Hill sediment samples were collected from six different sites around the Fildes Peninsula, each subject to varying levels of use pressure (Fig 1 and S1 Table).

The sampling sites S1, S5, and S6 were classified as non-impacted because they are remote, have limited human access, and lack roads. The details are as follows:

- **S1** is located southeast of the Great Wall Station, near a colony of *Pygoscelis papua* penguins, where four samples were collected (AB1, AA2, AD4, and AI1). Samples AB1, AA2, and AD4 are beach sediments collected along a transect, while AI1 was collected on a nearby hill 700 meters inland.

- **S5** is situated northeast of the Fildes Peninsula, where seven samples were collected (M1, N3, L3, O2, P2, P5, Q4). Samples M1, N3, and L3 were collected at Gradzinski Cove-Klotz Valley toward the Drake Passage, while samples Q4, P5, P2, and O2 were taken from the permafrost of the Collins Glacier. This site is inaccessible by car, and the nearest station (Artigas Base, Uruguay) is 4 km away.

- **S6** is highly isolated, accessible only by sea, and located northwest of the Fildes Peninsula, surrounded by the Collins Glacier. Four samples were collected from this site (H3, I2, J1, and J3).

Sampling sites S2, S3, and S4 are near the airport, research stations, and shelters, with consistent human activity, including scientific, logistical, and touristic operations. The details are as follows:

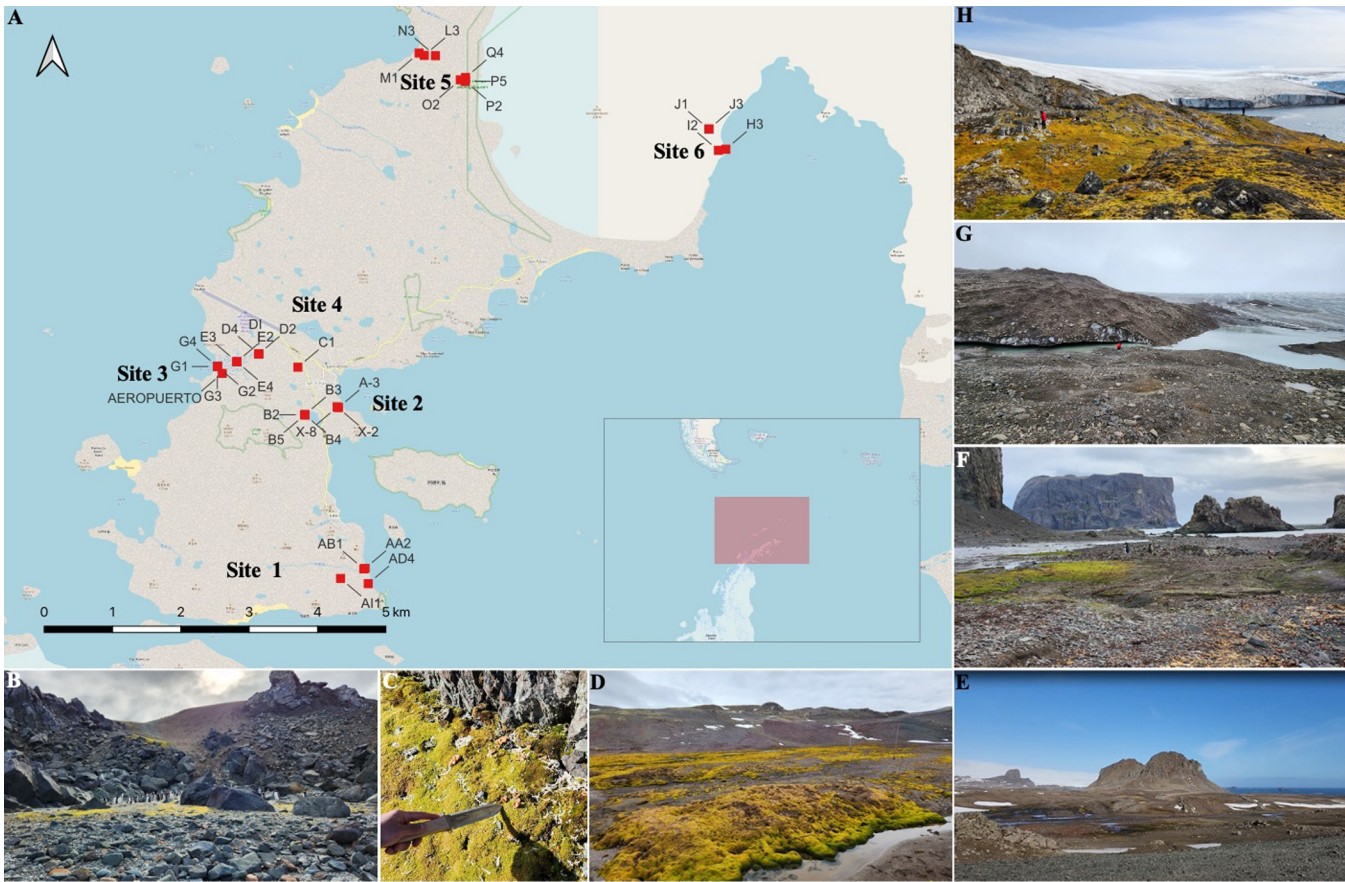

**Fig 1. Sites sampled during the Antarctic Scientific Expedition (ECA59), Fildes Peninsula, King George Island, Antarctica.** (A) Map showing collecting sites; (B) and (C) Pictures of sampling S1 (pristine), (D) Picture of sampling S2 (disturbed), (E) Picture of sampling S3 (disturbed), (F) Picture of sampling S4 (disturbed), (G) Picture of sampling S5 (pristine), near Bellinghausen Glacier, and (H) Picture of sampling S6 near to Bellinghausen Glacier (pristine). ©OpenStreetMap contributors (2015); maps retrieved in September 2024 from https://planet.openstreetmap.org.

- **S2** is located at the center of the Fildes Peninsula, 100 meters from Villa Las Estrellas and Julio Escudero Base. Anthropogenic influence, such as the movement of machinery for transporting supplies, equipment, and waste disposal, was observed at this site, which is close to Chilean Navy facilities, the Antarctic Institute, and the Russian Base. Seven samples were collected here (B2, B3, B4, B5, A3, X-2, and X-8).

- **S3** and **S4** are located in a valley adjacent to Teniente R. Marsh Airport, near a small creek. S3 has significant anthropogenic impact, including the presence of waste such as plastics, paper, wood, and metal. The transect at this site is 2 km long, beginning 200 meters from Villa Las Estrellas (C1), passing near the airport (D1, D2, D4), and ending 300 meters before the river mouth (E2, E3, E4).

- **S4** corresponds to the endpoint of the transect described for S3, with five samples collected at the river mouth (G1, G2, G3, G4, AP).

## Sample collection and processing

At each site, sediment samples were collected using a metal spatula, which was sterilized with alcohol after each use. Sterilized centrifuge Falcon tubes (28 mm x 120 mm) were used to

collect and transport the samples. Each sample was properly labeled, and the GPS location was recorded. All samples were stored at 4°C in the Julio Escudero Base Laboratories. They were then transported to the Laboratory of Applied and Sustainable Chemistry (LabQAS) at Universidad del Bío-Bío, Chile, shipped at 4°C to Dr. Leffler's lab at the University of Utah, USA and processed by Jonah Ventures (https://jonahventures.com/).

## DNA extraction and PCR amplification

Approximately 250 mg of sediment was placed into the extraction tube using sterilized tweezers within a laminar flow hood. The material was thoroughly lysed for 20 minutes using beads and a vortex mixer to break apart the tissue. Sample barcodes were recorded and assigned to a well within the 96-well plate or a numbered extraction tube. Plates or tubes were either immediately processed or stored at -20°C until the extraction process could be performed. Genomic DNA from the samples was extracted using the DNeasy 96 PowerSoil Pro Kit (384) (Cat # 47017), following the manufacturer's protocol. Genomic DNA was eluted into 100 μL and frozen at -20°C.

The internal transcribed spacer 1 (ITS1) region of the nuclear DNA was amplified by PCR using the ITS1-F (5'-CTTGGTCATTTAGAGGAAGTAA-3') and ITS2 (5'-GCTGCGTTCTTCAT CGATGC-3') primers [53,54]. Both forward and reverse primers also included a 5' adaptor sequence for subsequent indexing and Illumina sequencing. Each 25 μL PCR reaction was prepared according to the Promega PCR Master Mix specifications (Promega catalog # M5133, Madison, WI), consisting of 12.5 μL Master Mix, 0.5 μL of each primer, 1.0 μL of genomic DNA, and 10.5 μL of DNase/RNase-free H2O. DNA was amplified using the following conditions: initial denaturation at 95°C for 5 minutes, followed by 35 cycles of 45 seconds at 95°C, 1 minute at 50°C, and 90 seconds at 72°C, with a final elongation at 72°C for 10 minutes.

To determine amplicon size and PCR efficiency, each reaction was visually inspected using a 2% agarose gel, with 5 μL of each sample as input. Amplicons were then cleaned by incubating them with Exo1/SAP for 30 minutes at 37°C, followed by inactivation at 95°C for 5 minutes, and then stored at -20°C.

## PCR library and Illumina sequencing

A second round of PCR was performed to complete the sequencing library construction, appending the final Illumina sequencing adapters and integrating a sample-specific 12-nucleotide index sequence. The indexing PCR included Promega Master Mix, 0.5 μM of each primer, and 2 μL of template DNA (cleaned amplicon from the first PCR reaction). The PCR protocol consisted of an initial denaturation at 95°C for 3 minutes, followed by 8 cycles of 95°C for 30 seconds, 55°C for 30 seconds, and 72°C for 30 seconds.

Final indexed amplicons from each sample were cleaned and normalized using SequalPrep Normalization Plates (Life Technologies, Carlsbad, CA). A total of 25 μL of PCR amplicon was purified and normalized using the Life Technologies SequalPrep Normalization Kit (Cat# A10510-01), following the manufacturer's protocol. Samples were then pooled by adding 5 μL of each normalized sample to the pool.

Sample library pools were sent for sequencing on an Illumina MiSeq (San Diego, CA) at the Texas A&M AgriLife Genomics and Bioinformatics Sequencing Core facility, using the v2 500-cycle kit (Cat# MS-102-2003). Necessary quality control measures were conducted at the sequencing center prior to sequencing.

## Sequence data processing, taxonomic assignation, and distribution

Raw sequence data were demultiplexed using Pheniqs v2.1.0 [55], enforcing strict matching of sample barcode indices (i.e., no errors). Cutadapt v3.4 [56] was then used to remove adapters. For the ITS analysis pipeline, the DADA2 v1.32.0 package [57] was employed to generate filtered reads based on quality, length, and ambiguous nucleotides. After filtering, sequence data were denoised, merged to generate a dataframe without chimera sequences, along with sequence counts per sample. The resulting sequences were subjected to local alignment using BLAST [58] against QIIME files v10.0 [59], stored in the UNITE community database v10.0 [60]. Sequences with an identity percentage lower than 97% or query coverage lower than 60% were discarded and classified as "Unassigned," along with sequences without BLAST results. The remaining sequences were grouped into OTUs with 97% sequence similarity based on the taxonomic names provided in the QIIME files [61]. Finally, all taxa with less than 10% relative frequency in any sample were grouped as "Other." UpSet plots were generated to visualize read counts and genus counts, comparing site status (pristine or disturbed), using the UpSetR v1.4.0 package [62].

## Diversity, composition and ecological characterization of fungal communities from pristine and disturbed sites

Shannon and Simpson indices were calculated for each sample, yielding the average, standard deviation, maximum, and minimum values for both indices across sites using the vegan 2.6–4 package in R Studio, the diversity of the fungal samples was then assessed based on habitat (pristine or disturbed) and sampling sites. First, a global analysis of the data was performed, evaluating normality and variance homogeneity using the Shapiro-Wilk test and the Levene test (car package 3.1–2), respectively. The Shannon index was analysed based on habitat (pristine or disturbed) and sampling sites using an ANOVA test. Additionally, the post hoc Tukey test was applied, using the stats 4.3.1 package in R Studio, to evaluate pairwise differences between groups based on habitat and sampling sites.

The Simpson index, as it did not meet the assumptions of normality, was analysed using non-parametric tests. To compare groups by habitat, the Wilcoxon test from the stats 4.3.1 package in R Studio was used, while for analysis by sampling site, the Kruskal-Wallis test was applied. A subsequent post-hoc Dunn test from the FSA 0.9.5 package in R Studio was conducted to identify which pairs of groups showed statistically significant differences based on habitat and sampling sites. A more specific analysis was then carried out by creating subsets for the pristine and disturbed sampling sites. For each subset, the analyses of normality and variance homogeneity were repeated using the Shapiro-Wilk test and the Levene test, respectively, for each diversity index according to the sampling sites.

As the data from the subsets met the assumptions of normality, parametric tests were applied. To compare the diversity indices between the sites of each habitat, ANOVA was used, followed by a post-hoc Tukey test to identify significant differences between the sites within each habitat. No subsets were created for each sampling site to analyse each sampling zone individually, as the minimum required sample size for valid statistical analysis was not met. Furthermore, the available samples exhibited high homogeneity, which limited the ability to detect significant differences between sites. Finally, the results were visualised using boxplots of the diversity indices based on the habitat (status) of the sampling sites, with the ggplot 3.4.4 package in R Studio. The editing of the boxplot graphs and the relative abundance heatmap was carried out in Adobe Illustrator 28.7.3.

The relationship between the genus composition of the samples (using only genera with at least 0.1% relative abundance) and the pristine/disturbed site status was visualized using non-

metric multidimensional scaling (NMDS) based on Bray–Curtis distances in the vegan package v2.6–4 (k = 5) [63], and further represented through a heatmap created with the Complex-Heatmap R package v2.20.0 [64].

Additionally, fungal communities (at the genus level) were classified based on their feeding strategies into ecological guilds using the FUNGuild tool v1.1 and its associated fungal database (https://github.com/UMNFuN/FUNGuild) [65].

## Results

### Read counts and chimeric proportions vary across sites

We recovered an average of 12,749 paired-end reads from each sample, with a minimum length of 210 bp observed in 11 samples (ranging from 3,412 to 22,559 reads per sample; see S2 Table). A total of 154,179 reads (approximately 35.6% of the total number of reads) were removed due to short length (< 210 bp), low quality scores, or the presence of ambiguous nucleotides. In total, 54.06% of the paired-end reads were successfully merged, with a maximum length of 398 bp and an average length of 280.96 bp. Approximately 0.30% of the reads (an average of 21 chimeric sequences across 3 samples) were identified as chimeric non-specific sequences, which were subsequently removed from further analyses.

For each site, the diagrams show varying read counts. The site with the largest number of reads is J1, with more than 132 total reads, while the site with the fewest reads is E3, with only 8 total reads (S1 Fig). The largest intersection set is at site A-3, with 98 unique reads, followed by J1 with 53 reads. There are no common reads across all sites, and the largest set of site combinations, between B4 and B3, contains only 8 unique reads (see S1 Fig).

### Taxonomic coverage diversity, and relative abundance

The most abundant phyla detected were Ascomycota (14.66%), Chytridiomycota (9.82%), Basidiomycota (8.19%), Rozellomycota (7.07%), Glomeromycota (4.75%), and Mortierellomycota (4.00%), while 34.95% of the fungal diversity remained unidentified (Fig 2). Other minor phyla with abundance values ranging between 0.1% to 1% included Basidiolomycota, Monoblepharomycota, and Sanchytridiomycota. In total, 1,073 OTUs were obtained, corresponding to 114 fungal taxa (532 OTUs) at the genus level, while 541 OTUs remained unassigned. The assigned taxa were distributed across 59 orders, with Mortierellales from Mortierellomycota, Thelebolales, Helotiales, and Orbiliales from Ascomycota, Rhizophidiales from Chytridiomycota, Filobasidiales and Cystofilobasidiales from Basidiomycota, and Glomerales from Glomeromycota being the most abundant (Fig 3 and S3 Table).

### Linkages between fungal communities from pristine and disturbed sites

The distribution and number of genera for each site are shown in the UpSet diagram in Fig 4. The largest assemblage, with nearly 41 genera, was found at site A-3. Site A-3 also had 18 specific genera, followed by AA2 with 4 specific genera. *Betamyces* (Chytridiomycota) was detected at 29 of the 34 sites. *Mortierella* (Mortierellomycota) was found at five pristine sites and seven disturbed sites. The genus *Thelebolus* (Ascomycota) was the most predominant, occurring at 8 pristine and 7 disturbed sites. *Rhodotorula* (Basidiomycota) was present at one pristine site (J1) and four disturbed sites (A-3, B3, C1, and G1). Similarly, *Glomeraceae* gen. *incertae sedis* was more prevalent in disturbed sites. Across all sites, many fungi could not be assigned to any taxonomic rank and showed high relative abundance (Fig 5).

The fungal communities at sites AI1, G2, and C1 deviate from the dispersal trend observed at their respective sites (Fig 6). The lowest variance in fungal communities was found at sites 2,

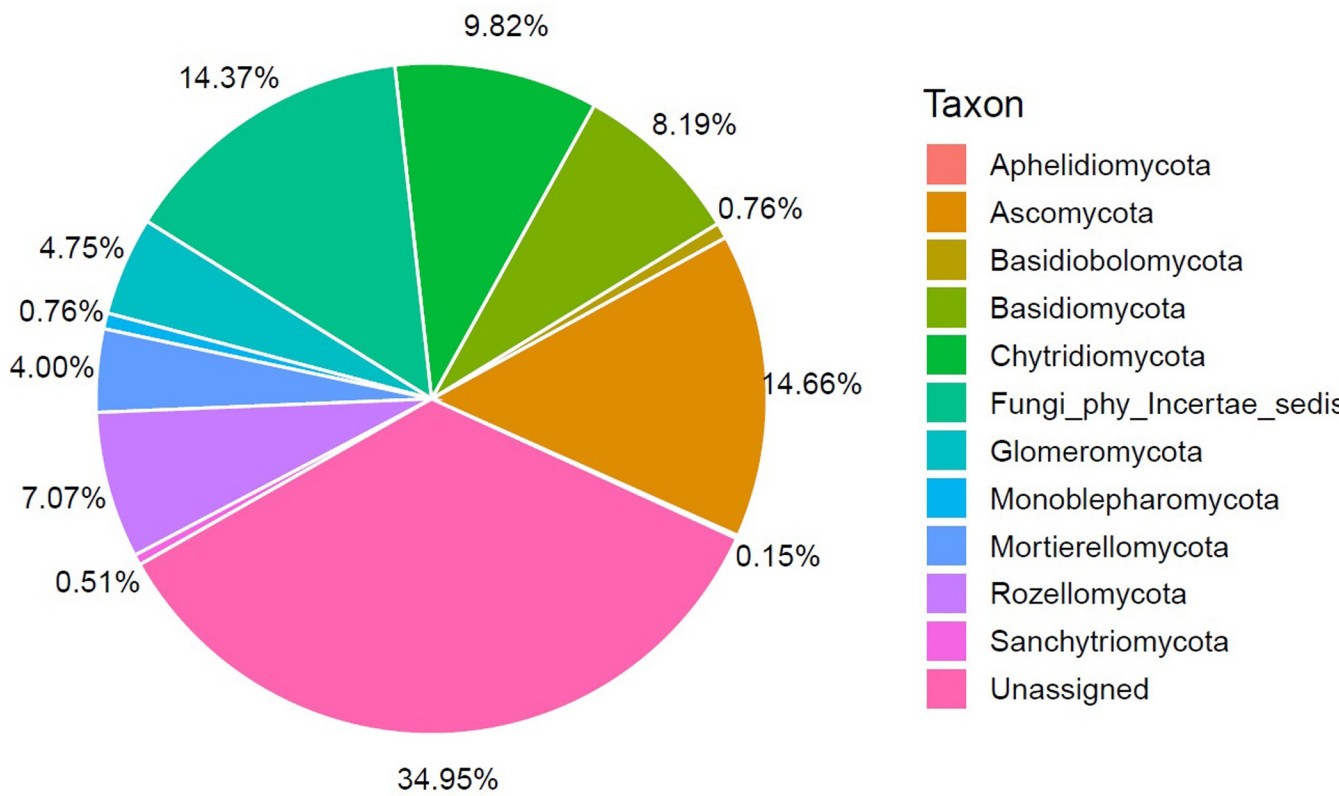

**Fig 2. Relative abundance of fungal communities at the phylum level from coastal sediment from ice-free areas on the Fildes Peninsula, King Jorge Island, Antarctica.** For the site locations, see Fig 1. Relative abundance values < 0.1% were discarded.

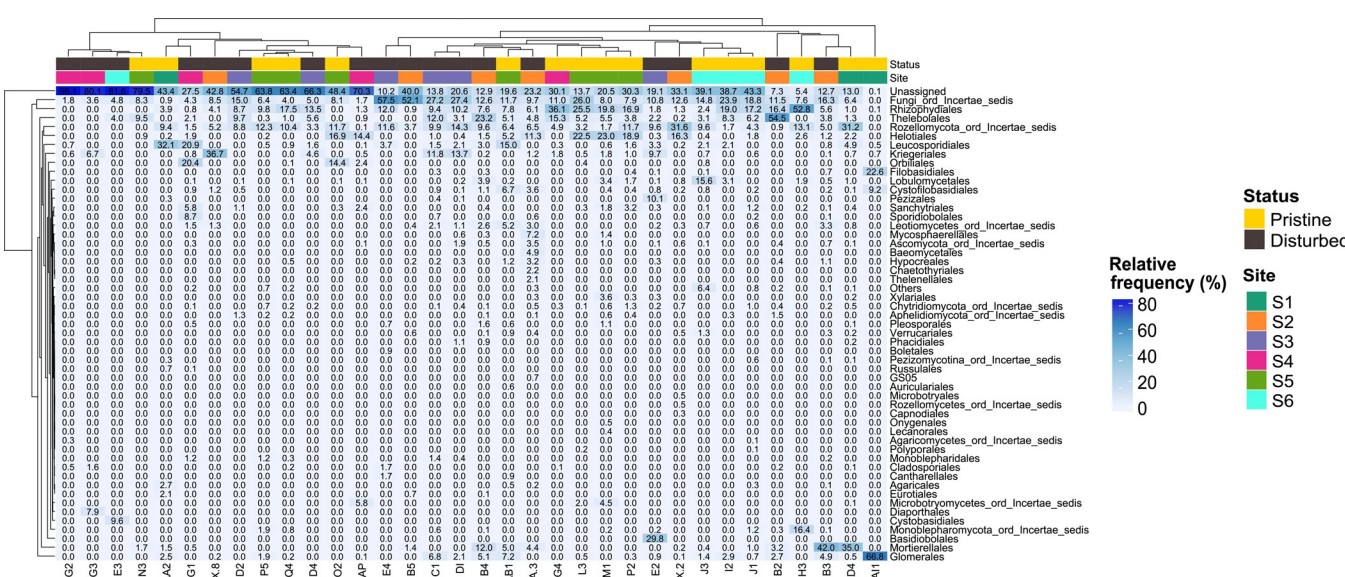

**Fig 3. Heatmap of the fungal communities at order level inhabiting coastal sediments from ice-free areas spanning pristine and disturbed sites in the Fildes Peninsula, King George Island, Antarctica.** Sites 1, 5, and 6 are pristine. Sites 2, 3, and 4 are disturbed. Relative abundance values < 0.1% were discarded.

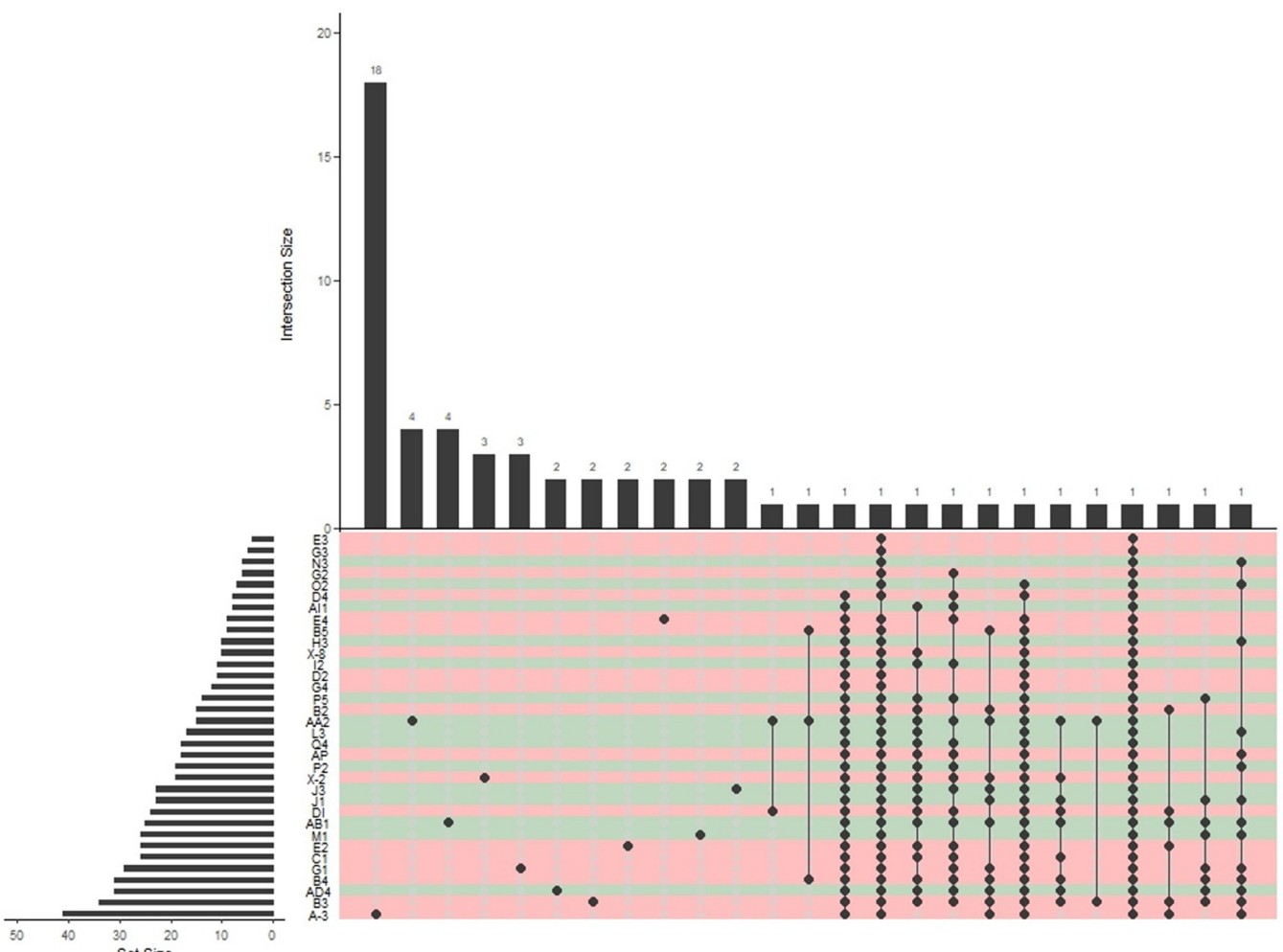

**Fig 4. Upset plot showing different number of genera across collecting sites in the Fildes Peninsula, King George Island, Antarctica.** Horizontal lines correspond to the total number of elements per sampling point and the various vertical lines represent the number of elements intercepted by the matrix. The black circles in the matrix indicate elements in common between the sampling points.

3, and 6. Furthermore, these communities did not differ significantly between pristine and disturbed sites.

There is a tendency for fungal communities to be less heterogeneous at the genus level in sites with anthropogenic impact compared to pristine sites (Fig 7). However, there is no significant difference in the minimum, maximum, or average values for either index, as inferred from the one-way ANOVA analyses. Among disturbed sites, sites 2 and 4 have a p-value closer to being significant in the Simpson index. The fungal community of sample A-3 is the most distinct compared to that of site 2. Similarly, the fungal communities at sites E3 and D4, as well as site B5 relative to site 2, deviate from the general trend of their respective sites.

The fungal taxa (genus level) were grouped into 23 different ecological guilds; the most abundant guilds were undefined saprotrophs, plant pathogens, plant saprotrophs, pollen saprotrophs, and endophytes (Fig 8 and S4 Table). No significant differences in the ecological guilds were detected between pristine and disturbed sites.

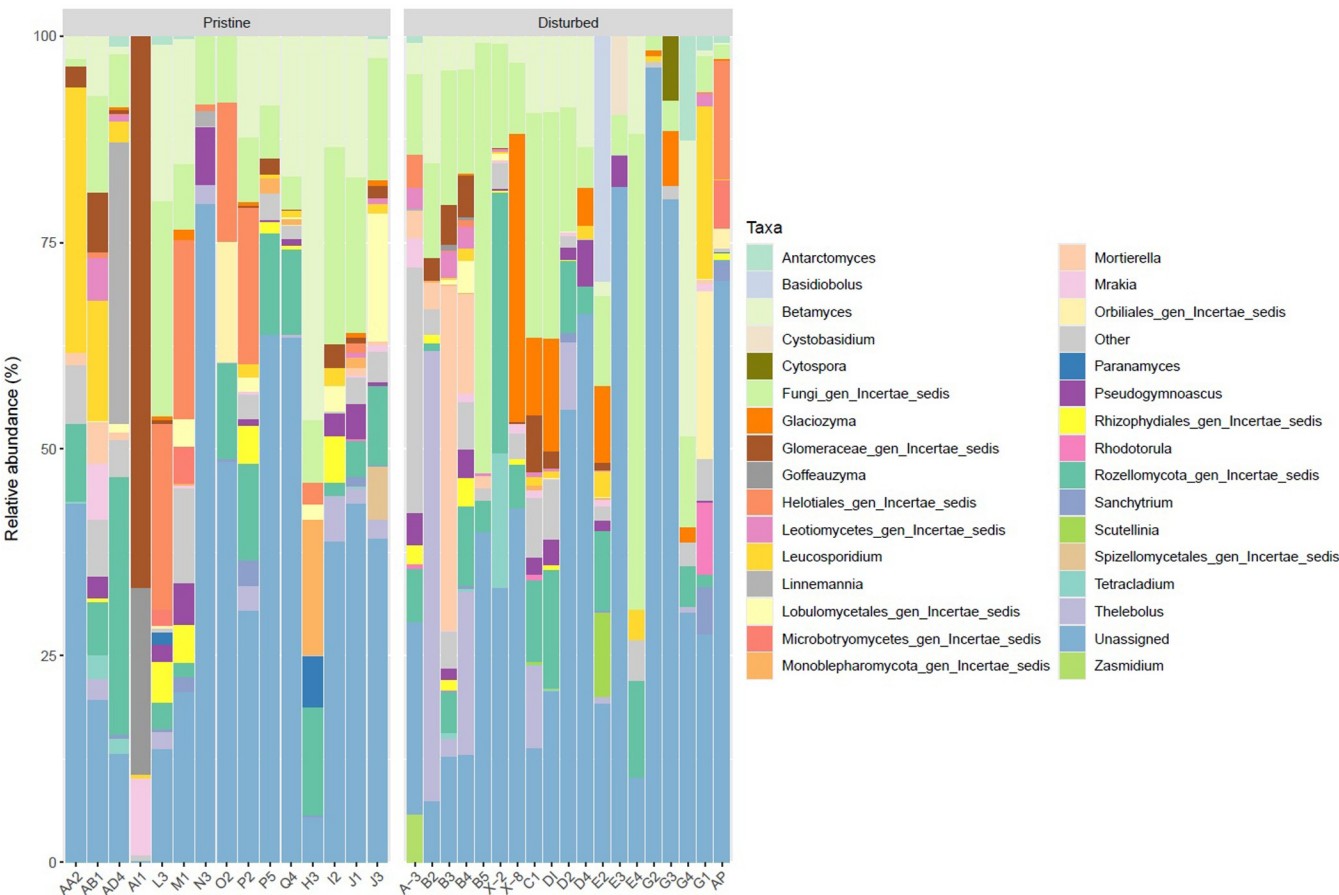

**Fig 5. Relative abundance of fungal communities at the genus level across coastal sediments from pristine and disturbed sites on the Fildes Peninsula, King George Island, Antarctica.** Fungal genera are ordered according to their mean relative abundances. All genera with less than 10% of relative frequency are grouped into 'Other'. Non taxonomically classified reads are named as 'Unassigned'.

## Discussion

Antarctic ice-free areas are critical ecological zones that support unique biodiversity, provide indicators of climate change, and offer valuable opportunities for scientific research. In recent years, studies on fungal communities in Antarctic ice-free areas using Illumina sequencing have provided valuable insights into the diversity, structure, and ecological roles of fungi in these extreme environments.

In this study, a total of 1,073 OTUs were distributed into 59 orders, corresponding to 114 known genera (532 OTUs). Some of these belong to more primitive, holocarpic, endobiotic fungi without cell walls, within the phylum Cryptomycota (Rozellomycota), which are parasites of various hosts, including Eumycota fungi and green algae [66]. These fungi have also been frequently reported in previous studies from both sediments and lake water [23,31–33,36,37], soils [6,40,43], rocks [67], marine sediments [12], and other diverse substrates [6,40] across various locations in Antarctica. Rojas-Jiménez et al. [31] indicated that the Cryptomycota were dominant in ice-covered lakes in the McMurdo Dry Valleys under minimal human influence, while Rosa et al. [40] reported Rozellomycota in soils from both protected and unprotected areas on Deception Island, Antarctica.

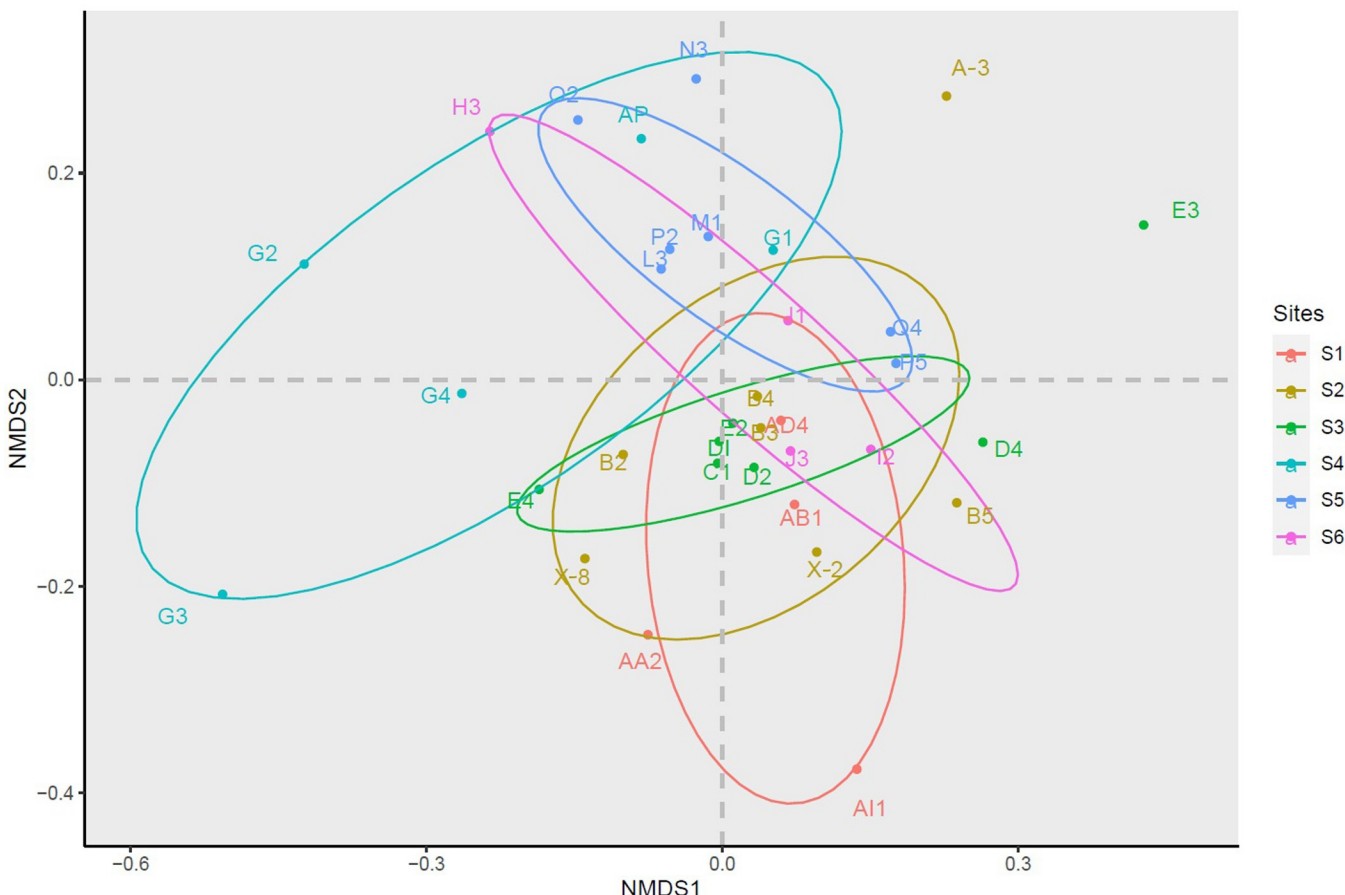

**Fig 6. NMDS plot of fungal OTUs inhabiting coastal sediments from ice-free areas across pristine and disturbed sites in the Fildes Peninsula, King George Island, Antarctica based on the ITS1 sequences.** For the site locations, see Fig 1. Dots represent locations: Sites 1, 5, and 6 are pristine and Sites 2, 3, and 4 are disturbed. Samples are grouped according to the collecting sites.

Chytridiomycota, which include forms with flagellated zoospores, are mostly saprotrophs, although they can also be parasites of animals, plants, and protists. Several previous studies have demonstrated the predominance of Chytridiomycota in Antarctic ecosystems. There are records of chytrids through DNA sequencing in lake sediments [33–38], soils [40,43,68], air and snow [40], marine sediments [13], lake water [31,33], and various substrates [6]. It has been suggested that parasitic chytrids significantly affect the food webs of freshwater bodies [69]. The genus *Betamyces* (Rhizophidiales) was the most widely distributed and abundant across sites with and without human intervention. This genus has also been reported by Gonçalves et al. [36,43] as dominant in lake sediments and soil samples from James Ross Island, Antarctica.

In this study, the genus *Mortierella*, which belongs to the family Mortierellomycota contains some psychrophilic species, was primarily found in disturbed sites. This genus has also been reported by Gonçalves et al. [43] from soils on James Ross Island and rocks from ice-free areas of the Union Glacier [30], as well as from the permafrost and active layers on Robert, Livingston, and Deception Islands by da Silva et al. [40], and in seasonal snow at Martel Inlet, King George Island, by Rosa et al. [45].

Ascomycota and Basidiomycota are the most dominant phyla across various substrates and locations in Antarctica. They have been reported in lake sediments [33–35,37,38], soils

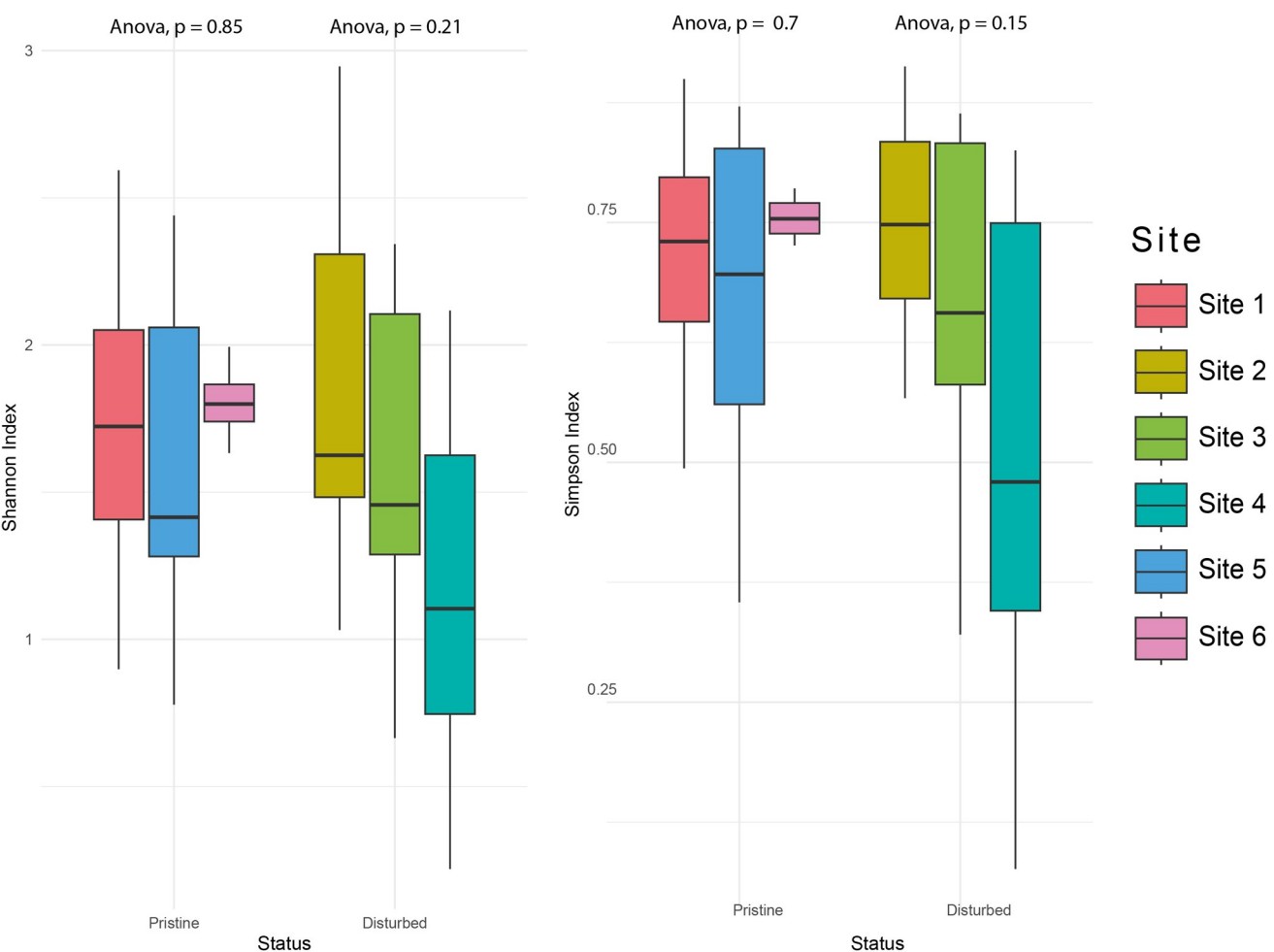

**Fig 7. Estimates of diversity indices of fungal communities in coastal sediments for each of the sites sampled on the Fildes Peninsula, King George Island, Antarctica.** (A) Shannon Index. (B) Simpson Index. Both diversity indices (97% sequence similarity) are delineated at the genus level.

[40,41,43], air and snow [44], marine sediments [13], lake water [31], and various substrates [6]. Similarly, in our study, Ascomycota was the most abundant phylum, while Basidiomycota ranked third in terms of abundance. Within Ascomycota, the orders Thelebolales, Helotiales, and Orbitales, and within Basidiomycota, Filobasidiales, and Cystofilobasidiales, were the most abundant. Some of these have already been reported as frequent in Antarctica in previous studies (e.g., [6,43]).

As in previous studies (e.g., [35,43,67]), the phyla Aphelidomycota, Basidiolomycota, and Monoblepharomycota were found to be rare. Arbuscular mycorrhizal fungi (AMF–Glomeromycota), widely distributed in terrestrial ecosystems, were abundant at one of the sampling sites (AI1). Newsham et al. [43] detected a low abundance of Glomeromycota in soils along a maritime Antarctic transect.

### Unassigned fungal diversity

In this study, one-third of the diversity could not be assigned to any known taxonomic group within the Eumycota fungi. Regardless of the site of origin and substrate in Antarctica, fungal

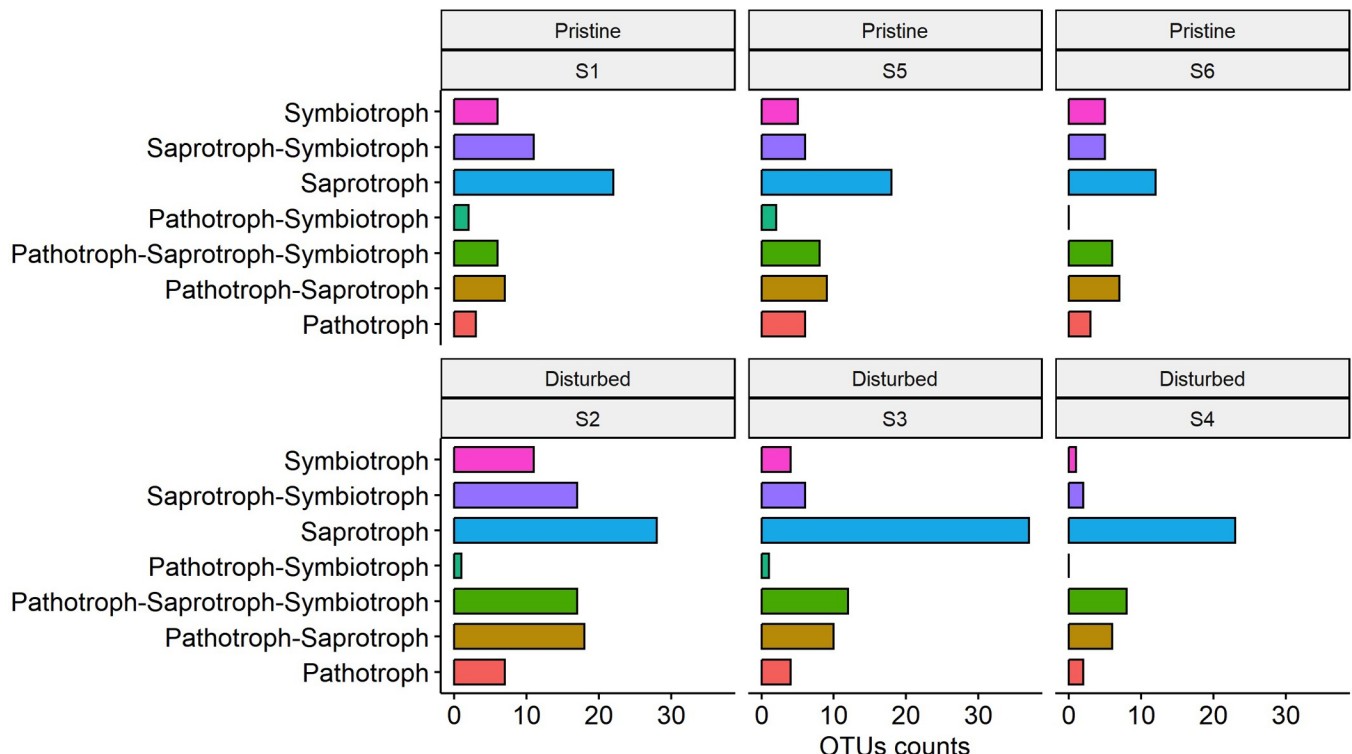

**Fig 8. Ecological associations of fungal communities inhabiting coastal sediments from ice-free areas in pristine and disturbed sites on the Fildes Peninsula, King George Island, Antarctica.** Samples from pristine and perturbed sites are depicted. The fungal taxa (OTUs) were submitted ecological guilds with the FUNGuild tool and the corresponding database. Guild output was distributed into individual guilds.

DNA metabarcoding could not be taxonomically assigned or could be only assigned at higher taxonomic levels, and some of the reported values include 43 (66%) ASVs [34], 35 (39.7%) ASVs [43], more than 50% of OTU reads [41], 55.4% ASVs [13], 106 (38.4%) OTUs [68], 78 ASVs [44], 40 ASVs [37], 29 ASVs [45], 95 (47.57%) ASVs [35], 26.7% ASVs [33], and 9,892 ASVs [6]. These results suggest that databases do not adequately represent these taxa, or that there is a large diversity yet to be identified in these habitats. Therefore, there is a need to increase taxonomic research in Antarctica to expand our knowledge of existing fungal species and better understand the function and diversity of these microbial communities.

### Ecological functionality

In general, we know very little about the functional ecology of Antarctic fungi. The ecological guilds of the fungal communities detected here include undefined saprotrophs, followed by plant pathogens, plant saprotrophs, pollen saprotrophs, and endophytes. This is consistent with previous studies investigating the ecological functions of fungi in various Antarctic environments, indicating that saprotrophs are the dominant functional group, followed by plant and animal pathogens, and symbionts (e.g., [6,13,34,35,37,40,45]).

The most prevalent fungal guild identified in this study was that of the saprotrophs. In soils, saprotrophic fungi play an important ecological role in the decomposition of organic matter and nutrient cycling. Previous studies have revealed the presence of pathogenic fungi in marine sediments [13], which may have been introduced by human activities, as suggested by Rosa et al. [40]. In this study, the cosmopolitan genus *Rhodotorula* was mainly detected in

disturbed sites. This genus has been reported in snow [14], air and snow [44], and marine sediments [29].

## Impact of human activities on fungal communities

Although our analyses indicate an impact on the diversity and composition of fungal communities due to human activities, this was not statistically significant. We interpret these results as a sign that anthropogenic influence is neither intense nor prolonged, and that the short distance between sites could contribute to similar assemblages among fungal communities. Although we detected several genera with global distributions, our ITS1-based approach did not allow species-level identification. This prevents us from determining whether they are introduced or endemic species. In the study by Rosa et al. [40], based on ITS2, the diversity of soil fungi in protected and non-protected areas on Deception Island was analyzed, suggesting that some "rare" fungal taxa could have been transported by human activities. Unlike these authors, we did not detect genera of fungi pathogenic to humans or other animals.

So far, a little over 1,000 fungal species have been recorded in Antarctica [70]. However, considering the wide fungal diversity reported in recent metabarcoding studies, this likely represents only the "tip of the iceberg" in terms of true diversity. Future mycological campaigns should include the collection of various substrates for the isolation of fungi in culture and fruiting bodies. Additionally, detailed morphological analyses, sequencing, and taxonomic studies are needed to describe and identify species, determining whether they correspond to already described species or new ones.

## Conclusion

Illumina sequencing revealed a high diversity and ecological functionality of fungal communities inhabiting sediments from ice-free areas in both pristine and human-disturbed sites on the Fildes Peninsula, Isla Rey Jorge, Antarctica. It was found that human activities do not have a significant impact on the fungal community composition in these ice-free areas, thus rejecting our working hypothesis. We interpret these findings as indicating that the impact may be either not intense or prolonged, the spatial distance between the sampled sites may be too small, and/or the environmental factors may be similar across sites.

Considering that nearly one-third of the diversity could not be assigned to any specific taxonomic category, it is essential to first identify the species in these extreme habitats before initiating any protection and conservation strategies. The large proportion of unidentified fungal sequences might indicate a high level of endemism in Antarctica. Therefore, as we still know very little about the true diversity, composition, and ecology of fungal communities in these habitats, further taxonomic studies, including culture-based approaches and DNA barcoding, are required to enhance our understanding of the species inhabiting ice-free areas.

## Supporting information

**S1 Table. List of the samples sequenced in this study ordered by collection site in the Fildes Península, Isla Rey Jorge, Antarctica.** The coordinates of each sample are provided, with samples categorized as either pristine or disturbed based on the level of human impact at the sampled sites.
(XLSX)

**S2 Table. Summary of data sequencing of samples collected across collecting sites in the Fildes Península, Isla Rey Jorge, Antarctica.** The code of collection site, number raw reads,

the number of reads after filtering and assembled reads are given.
(XLSX)

**S3 Table. Distribution and relative abundance of the fungal communities at phylum level from samples collected in the Fildes Península, Isla Rey Jorge, Antarctica.** The collection sites are given.
(XLSX)

**S4 Table. Functional guilds of the fungal communities at genus level (OTU) from samples collected in the Fildes Península, Isla Rey Jorge, Antarctica.** The collection sites are given.
(XLSX)

**S1 Fig. Upset plot showing different numbers of reads across collecting sites in the Fildes Península, Isla Rey Jorge, Antarctica.** Relationships between reads and sediment samples collected at sites with and without anthropogenic impact are shown.
(PNG)

## Acknowledgments

We thank B. Romero for helping with the statistical analyses and C. Riquelme for designing the map, as well as several pre- and postgraduate students for processing samples in our labs. We also thank two anonymous reviewers for the useful comments, which have helped to improve our manuscript.

## Author Contributions

**Conceptualization:** Sigisfredo Garnica, Jaime R. Cabrera-Pardo.

**Data curation:** Genaro Soto-Rauch, Christian Núñez, Jonhatan Gómez-Espinoza, Enzo Romero, Ramón Ahumada-Rudolph.

**Formal analysis:** Genaro Soto-Rauch.

**Funding acquisition:** Sigisfredo Garnica, Ellen M. Leffler, Jaime R. Cabrera-Pardo.

**Investigation:** Christian Núñez, Jonhatan Gómez-Espinoza, Enzo Romero, Ramón Ahumada-Rudolph.

**Methodology:** Sigisfredo Garnica, Genaro Soto-Rauch, Christian Núñez, Jonhatan Gómez-Espinoza, Enzo Romero, Ramón Ahumada-Rudolph.

**Supervision:** Jaime R. Cabrera-Pardo.

**Writing – original draft:** Sigisfredo Garnica, Jaime R. Cabrera-Pardo.

**Writing – review & editing:** Ellen M. Leffler, Jaime R. Cabrera-Pardo.

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
