## [Decision Letter · Decision Letter 0]

9 Dec 2024

PONE-D-24-48625High diversity of fungal ecological groups from ice-free pristine and disturbed areas in the Fildes Peninsula, King George Island, AntarcticaPLOS ONE

Dear Dr. Garnica,

Thank you for submitting your manuscript to PLOS ONE. After careful consideration, we feel that it has merit but does not fully meet PLOS ONE’s publication criteria as it currently stands. Therefore, we invite you to submit a revised version of the manuscript that addresses the points raised during the review process.

We look forward to receiving your revised manuscript.

Kind regards,

Erika Kothe

Academic Editor

PLOS ONE

2. Thank you for stating the following financial disclosure: [Instituto Antártico Chileno (INACH), grant INACH RT_16-21. J.R.C-P and VRIP - Universidad del Bio Bio, grants GI2310643 and EQ2326450.]. Please state what role the funders took in the study. If the funders had no role, please state: "The funders had no role in study design, data collection and analysis, decision to publish, or preparation of the manuscript." If this statement is not correct you must amend it as needed. Please include this amended Role of Funder statement in your cover letter; we will change the online submission form on your behalf.

3. For studies involving third-party data, we encourage authors to share any data specific to their analyses that they can legally distribute. PLOS recognizes, however, that authors may be using third-party data they do not have the rights to share. When third-party data cannot be publicly shared, authors must provide all information necessary for interested researchers to apply to gain access to the data. (https://journals.plos.org/plosone/s/data-availability#loc-acceptable-data-access-restrictions) For any third-party data that the authors cannot legally distribute, they should include the following information in their Data Availability Statement upon submission: 1) A description of the data set and the third-party source 2) If applicable, verification of permission to use the data set 3) Confirmation of whether the authors received any special privileges in accessing the data that other researchers would not have 4) All necessary contact information others would need to apply to gain access to the data.

6. Please ensure that you refer to Figure 1 and 3 in your text as, if accepted, production will need this reference to link the reader to the figure.

7. Please include a caption for figure 1, 2, 4, 5 and 8

8. Please remove your figures from within your manuscript file, leaving only the individual TIFF/EPS image files, uploaded separately. These will be automatically included in the reviewers’ PDF**.**

Additional Editor Comments:

The expert reviewers have recommended some changes that should be carefully considered with a minor revision.

Reviewers' comments:

Reviewer's Responses to Questions

**Comments to the Author**

1. Is the manuscript technically sound, and do the data support the conclusions?

Reviewer #1: Yes

Reviewer #2: Yes

2. Has the statistical analysis been performed appropriately and rigorously? 

Reviewer #1: Yes

Reviewer #2: Yes

3. Have the authors made all data underlying the findings in their manuscript fully available?

Reviewer #1: Yes

Reviewer #2: Yes

4. Is the manuscript presented in an intelligible fashion and written in standard English?

Reviewer #1: Yes

Reviewer #2: Yes

5. Review Comments to the Author

Reviewer #1: 

The manuscript presents highly relevant results that contribute significantly to scientific understanding, offering novel and important insights into the ecosystem of the Antarctic continent. However, I have a few suggestions for improvement.

Introduction Text:

• Line 52-53: I recommend including relevant references regarding endemic fungi in Antarctica to support the discussion and provide a broader context for the research.

• Line 61: The word "fishing" should be removed, as it may not be directly relevant to the topic.

• Line 64: It would be beneficial to include citations for the studies referenced in this sentence to provide proper attribution and strengthen the credibility of the information presented.

Methodology text

• It would be helpful to provide a clear explanation of what is meant by "sediment" in the context of your study. (part of soil?????)

Results:

• Figure 2: The Aphelidiomycota phylum is not represented in the figure. I recommend either updating the figure to include this group or providing an explanation in the text regarding its omission or its absence from the group.

• Methodology and Discussion: The authors mention the Shannon and Simpson indices in the methodology, but these indices are not discussed in the results or the discussion. It would be useful to either include a discussion of these indices in the appropriate section or revise the methodology to remove the mention if they were not ultimately used in the analysis.

The recommendation is for minor revisions to the manuscript. The suggested changes are relatively straightforward and involve clarifying certain points, adding relevant citations, and ensuring consistency between the methodology, results, and discussion sections. These adjustments will enhance the clarity and overall quality of the manuscript.

Reviewer #2: In this study, the authors present a metabarcoding analysis of fungal diversity in the Antartica comparing pristine sites with disturbed sites. The work include the commonly used analysis for community ecology. While the importance of the study relies on the system and is mainly descriptive, the paper is a valuable contribution to the study fungal diversity.

In general, the manuscript is well written, the objectives are clear, and the statistical analyses basic, but adequate for the objectives.

Comments

Consider to represent differences among sites using a dissimilarity measure that uses only presence/absence of genera/OTUS.

Specify the version of the UNITE database.

Line 319. I’m not sure that the genus Mortierella should be considered as psychrophilic. The genus as a wide distribution and can live as saprotrophs, live in fecal pellets or even as pathogens. Do you refer to any species in particular? It is important to differentiate “strict” from “facultative”.

Line 320. The reference Gonçalves et al is cited in References as 43 not 42

Line 342. Here, the authors mention ASVs, however they were not defined previously. So, in this case, the ASVs were first obtained and then grouped as OTUS? Please describe in methods.

Please indicate in Figure 1 the Pristine and the Disturbed locations.

Figure 3. Please change the color palette of the relative frequency to make easier to evaluate differences across sites.

Figure 6. Specify which sites are from Pristine and the Disturbed locations. Also, dots represent locations, not OTUs.

Figure 7. The legends should be rescaled to distinguish the boxplot colors. The differences reported are within Pristine and Disturbed groups? What about between groups? In the figure, the pink box may be statistically to the light-blue one, ¿did the authors consider to use a ad-hoc Tukey test? On the other hand, ¿ANOVA is suitable for this kind of data? A test for variance homogeneity should be performed prior to the test.

Figure 8. Use relative frequency or other standardized measure to avoid bias due differences in sample sizes.

6. PLOS authors have the option to publish the peer review history of their article (what does this mean?). If published, this will include your full peer review and any attached files.

Reviewer #1: No

Reviewer #2: No

---

## [Author Response · Author response to Decision Letter 0]

18 Dec 2024

ONE-D-24-48625

High diversity of fungal ecological groups from ice-free pristine and disturbed areas in the Fildes Peninsula, King George Island, Antarctica

PLOS ONE

We would like to thank the anonymous reviewers and editor for their thorough and helpful comments, which substantially improved the revised version of our manuscript. In the following pages, we give the details of all changes/improvements to the revised version of our manuscript.

We have improved our manuscript (incl. figures) following the reviewers’ recommendations/criticisms (Please see “Point-by-point response to reviewers”). 

Sincerely,

Sigisfredo Garnica

Point-by-point response to reviewers.

Dear Dr. Garnica,

Thank you for submitting your manuscript to PLOS ONE. After careful consideration, we feel that it has merit but does not fully meet PLOS ONE’s publication criteria as it currently stands. Therefore, we invite you to submit a revised version of the manuscript that addresses the points raised during the review process.

We look forward to receiving your revised manuscript.

Kind regards,

Erika Kothe

Academic Editor

PLOS ONE

We have revised our manukript to meet PLOS ONE's style requirements, including file naming.

2. Thank you for stating the following financial disclosure: [Instituto Antártico Chileno (INACH), grant INACH RT_16-21. J.R.C-P and VRIP - Universidad del Bio Bio, grants GI2310643 and EQ2326450.]. Please state what role the funders took in the study. If the funders had no role, please state: "The funders had no role in study design, data collection and analysis, decision to publish, or preparation of the manuscript." If this statement is not correct you must amend it as needed. Please include this amended Role of Funder statement in your cover letter; we will change the online submission form on your behalf.

Funders provided the resources for Antarctic expeditions and laboratory expenses to perform the experiments indicated in the study.

3. For studies involving third-party data, we encourage authors to share any data specific to their analyses that they can legally distribute. PLOS recognizes, however, that authors may be using third-party data they do not have the rights to share. When third-party data cannot be publicly shared, authors must provide all information necessary for interested researchers to apply to gain access to the data. (https://journals.plos.org/plosone/s/data-availability#loc-acceptable-data-access-restrictions) For any third-party data that the authors cannot legally distribute, they should include the following information in their Data Availability Statement upon submission: 1) A description of the data set and the third-party source 2) If applicable, verification of permission to use the data set 3) Confirmation of whether the authors received any special privileges in accessing the data that other researchers would not have 4) All necessary contact information others would need to apply to gain access to the data.

Our study uses our own data that has not been previously published anywhere.

The ORCID iD for the corresponding author was validated in Editorial Manager.

We use map images from OpenStreetMap. Now we include proper attribution to the source in our Figure caption.

6. Please ensure that you refer to Figure 1 and 3 in your text as, if accepted, production will need this reference to link the reader to the figure.

Revised.

7. Please include a caption for figure 1, 2, 4, 5 and 8

The mentioned figures have captions.

8. Please remove your figures from within your manuscript file, leaving only the individual TIFF/EPS image files, uploaded separately. These will be automatically included in the reviewers’ PDF.

The figures were removed from within the manuscript file.

The reference list was checked.

Additional Editor Comments:

The expert reviewers have recommended some changes that should be carefully considered with a minor revision.

Reviewers' comments:

Reviewer's Responses to Questions

Comments to the Author

1. Is the manuscript technically sound, and do the data support the conclusions?

Reviewer #1: Yes

Reviewer #2: Yes

2. Has the statistical analysis been performed appropriately and rigorously? 

Reviewer #1: Yes

Reviewer #2: Yes

3. Have the authors made all data underlying the findings in their manuscript fully available?

Reviewer #1: Yes

Reviewer #2: Yes

4. Is the manuscript presented in an intelligible fashion and written in standard English?

Reviewer #1: Yes

Reviewer #2: Yes

5. Review Comments to the Author

Reviewer #1: 

The manuscript presents highly relevant results that contribute significantly to scientific understanding, offering novel and important insights into the ecosystem of the Antarctic continent. However, I have a few suggestions for improvement.

Introduction Text:

• Line 52-53: I recommend including relevant references regarding endemic fungi in Antarctica to support the discussion and provide a broader context for the research.

Some relevant references were included.

• Line 61: The word "fishing" should be removed, as it may not be directly relevant to the topic.

Removed

• Line 64: It would be beneficial to include citations for the studies referenced in this sentence To provide proper attribution and strengthen the credibility of the information presented.

Citation was included

Methodology text

• It would be helpful to provide a clear explanation of what is meant by "sediment" in the context of your study. (part of soil?????)

The term sediment as used in our manuscript correspond to glacial Hill sediment. This information was included in Material & Methods.

Results:

• Figure 2: The Aphelidiomycota phylum is not represented in the figure. I recommend either updating the figure to include this group or providing an explanation in the text regarding its omission or its absence from the group.

The reviewer is correct; there are taxa with less than 0.1% relative abundance, and they are not included in the figure. Based on this principle, we have removed "the Aphelidiomycota" from the text.

• Methodology and Discussion: The authors mention the Shannon and Simpson indices in the methodology, but these indices are not discussed in the results or the discussion. It would be useful to either include a discussion of these indices in the appropriate section or revise the methodology to remove the mention if they were not ultimately used in the analysis.

The main results from both the Shannon and Simpson indices are described on lines 310 to 316 and discussed on lines 410 to 413, respectively.

The recommendation is for minor revisions to the manuscript. The suggested changes are relatively straightforward and involve clarifying certain points, adding relevant citations, and ensuring consistency between the methodology, results, and discussion sections. These adjustments will enhance the clarity and overall quality of the manuscript.

Reviewer #2: In this study, the authors present a metabarcoding analysis of fungal diversity in the Antartica comparing pristine sites with disturbed sites. The work include the commonly used analysis for community ecology. While the importance of the study relies on the system and is mainly descriptive, the paper is a valuable contribution to the study fungal diversity.

In general, the manuscript is well written, the objectives are clear, and the statistical analyses basic, but adequate for the objectives.

Comments

Consider to represent differences among sites using a dissimilarity measure that uses only presence/absence of genera/OTUS.

Fungal taxa (OTUs) are used to determine ecological guilds, rather than other assignations.

Specify the version of the UNITE database.

The version of the UNITE database was included

Line 319. I’m not sure that the genus Mortierella should be considered as psychrophilic. The genus as a wide distribution and can live as saprotrophs, live in fecal pellets or even as pathogens. Do you refer to any species in particular? It is

---

## [Decision Letter · Decision Letter 1]

2 Jan 2025

High diversity of fungal ecological groups from ice-free pristine and disturbed areas in the Fildes Peninsula, King George Island, Antarctica

PONE-D-24-48625R1

Dear Dr. Garnica,

We’re pleased to inform you that your manuscript has been judged scientifically suitable for publication and will be formally accepted for publication once it meets all outstanding technical requirements.

Kind regards,

Erika Kothe

Academic Editor

PLOS ONE

Additional Editor Comments (optional):

Reviewers' comments:

Reviewer's Responses to Questions

**Comments to the Author**

1. If the authors have adequately addressed your comments raised in a previous round of review and you feel that this manuscript is now acceptable for publication, you may indicate that here to bypass the “Comments to the Author” section, enter your conflict of interest statement in the “Confidential to Editor” section, and submit your "Accept" recommendation.

Reviewer #2: All comments have been addressed

2. Is the manuscript technically sound, and do the data support the conclusions?

Reviewer #2: Yes

3. Has the statistical analysis been performed appropriately and rigorously? 

Reviewer #2: Yes

4. Have the authors made all data underlying the findings in their manuscript fully available?

Reviewer #2: Yes

5. Is the manuscript presented in an intelligible fashion and written in standard English?

Reviewer #2: Yes

6. Review Comments to the Author

Reviewer #2: Comments were responded accordingly.

Please, check that images resolution meets Plos One standards.

7. PLOS authors have the option to publish the peer review history of their article (what does this mean?). If published, this will include your full peer review and any attached files.

Reviewer #2: No

---

## [Editor Report · Acceptance letter]

10 Jan 2025

PONE-D-24-48625R1 

PLOS ONE

Dear Dr. Garnica, 

I'm pleased to inform you that your manuscript has been deemed suitable for publication in PLOS ONE. Congratulations! Your manuscript is now being handed over to our production team.

Kind regards, 

on behalf of

Prof. Dr. Erika Kothe 

Academic Editor

PLOS ONE